# Pathogenic *BRCA* Variants as Biomarkers for Risk in Prostate Cancer

**DOI:** 10.3390/cancers13225697

**Published:** 2021-11-14

**Authors:** Ciara S. McNevin, Karen Cadoo, Anne-Marie Baird, Pierre Murchan, Orla Sheils, Ray McDermott, Stephen Finn

**Affiliations:** 1Department of Histopathology and Morbid Anatomy, Trinity Translational Medicine Institute, Trinity College Dublin, D08 W9RT Dublin, Ireland; MCNEVINC@tcd.ie (C.S.M.); murchanp@tcd.ie (P.M.); 2Department of Medical Oncology, St. James Hospital, D08 NHY1 Dublin, Ireland; kcadoo@stjames.ie; 3School of Medicine, Trinity Translational Medicine Institute, St. James Hospital, D08 W9RT Dublin, Ireland; bairda@tcd.ie (A.-M.B.); osheils@tcd.ie (O.S.); 4Science Foundation Ireland Centre for Research Training in Genomics Data Science, School of Mathematics, Statistics and Applied Mathematics, National University of Ireland, H91 TK33 Galway, Ireland; 5Department of Medical Oncology, Tallaght University Hospital, D24 NR0A Dublin, Ireland; ray.mcdermott@tuh.ie; 6Department of Medical Oncology, St. Vincent’s University Hospital, D04 YN26 Dublin, Ireland

**Keywords:** *BRCA2*, *BRCA1*, biomarker, prostate cancer, gene mutation, screening, treatment strategies

## Abstract

**Simple Summary:**

Historically, the treatment of prostate cancer was a blanket approach for all. Prostate cancer has not benefitted from targeted treatments based on specific tumour characteristics (ie. Particular genetic or molecular patterns) the way other cancers have. This is important as studies have shown that prostate cancer patients with certain errors in their genes, such as BRCA2 or BRCA1, are more likely to have worse disease and poorer outcome. These patients can be treated successfully with a group of drugs called ‘PARP inhibitors’. This paper examines the prognostic, clinical and therapeutic role of *BRCA2/BRCA1* mutations across the evolution of PCa. The impact of the inclusion of *BRCA* genes on genetic screening will also be outlined.

**Abstract:**

Studies have demonstrated that men with Prostate Cancer (PCa) harboring *BRCA2/BRCA1* genetic aberrations, are more likely to have worse disease and a poorer prognosis. A mutation in *BRCA2* is known to confer the highest risk of PCa for men (8.6 fold in men ≤65 years) making *BRCA* genes a conceivable genomic biomarker for risk in PCa. These genes have attracted a lot of research attention however their role in the clinical assessment and treatment of PCa remains complex. Multiple studies have been published examining the relationship between prostate cancer and *BRCA* mutations. Here *BRCA* mutations are explored specifically as a biomarker for risk in PCa. It is in this context, we examined the prognostic, clinical and therapeutic role of *BRCA2/BRCA1* mutations across the evolution of PCa. The impact of the inclusion of *BRCA* genes on genetic screening will also be outlined.

## 1. Introduction

Prostate Cancer (PCa) was the second most common cancer diagnosis made in men (14.1%) and the fifth leading cause of death (6.8%) worldwide in 2020 [1]. Population expansion and improved life expectancies across the globe are set to contribute to an increase in PCa [2] rendering it an major worldwide health concern. While there has been an emergence of novel treatments in the last ten years, even now PCa is a major source of cancer deaths in males [3]. Older age is the chief risk factor with greater than three quarters of all PCa detection made in men over the age of 65 years [4]. Family history and genetic predisposition such as pathogenic variants *BRCA1/BRCA2* have also been identified as important risk factors [5,6]. Other genes important for DNA repair can also play a role if mutated (see Table 1). These extend to the DNA mismatch repair genes *MLH1*, *MSH2, PMS2, MSH6,* and *EPCAM* [7] that play a role in Lynch syndrome. When functioning correctly *BRCA1/BRCA2* are important tumor suppressor genes with multiple functions including transcription and complex cell cycle control [8]. However, their primary role is in the repair of DNA double stranded breaks (DSB) through the initiation of homologous recombination (HR). Failure of this system due to mutations in these genes can predispose an individual to malignancies such as breast, ovarian, pancreatic and prostate cancers [9]. A mutation in *BRCA2* is known to confer the highest risk of PCa for men (8.6 fold in men ≤65 years), with *BRCA1* demonstrating increased risk, albeit to a lesser extent (3.5 fold) [10]. These genes have attracted a lot of research attention however their role in the clinical assessment and treatment of PCa remains complex. This review seeks to address the role of *BRCA2* and *BRCA1* mutations in PCa in terms of the clinical and therapeutic implications starting with their discovery in the 1990s.

## 2. Evolution of Knowledge of *BRCA* Genes Risk in Prostate Cancer

Dr. Mary-Claire King was the first to demonstrate that a single gene on chromosome 17q21 (then named *BRCA1*) was responsible for breast and ovarian cancer in many families in the 1990s, thus demonstrating a heredity component in these cancers and proving that gene mutations could predict vulnerability to these diseases [12]. Her discovery revolutionized the study of cancer genetics and has proved seminal in investigating heritability of many diseases and cancer subtypes [13]. The impact of germline mutations of *BRCA1* and *BRCA2* in breast and ovarian cancer are now well defined. *BRCA1* and *BRCA2* mutations result in 70% risk of developing breast cancer by the age of 80 [14], compared to 12% of the unaffected population. These genes impact early diagnosis, prevention and therapeutics in breast cancer, and have led to the refinement of national and international genetic screening guidelines. The clinical impact of the role of DNA damage repair genes is still evolving in PCa, although it likely mirrors the path of hereditary breast and ovarian cancer [13]. Indeed much of the initial observations on PCa heritability arose via a circuitous route from breast cancer studies examining *BRCA1* and *BRCA2*. The recognized association of breast cancer with PCa in families has been reported since the 1970s [15]. Important Icelandic studies in the 1990s reported that male relatives in breast cancer families were noted to have a 2–3 fold risk of PCa [16]. Further studies showed that males who harbored a germline *BRCA1* and *BRCA2* mutation, identified via a family history of breast cancer, were reported to have an increased the risk of PCa by three-fold and seven-fold, respectively [17,18]. Therefore, from breast cancer studies, it was first derived that men with *BRCA1* and *BRCA2* mutations were at higher risk for PCa.

## 3. Family History as a Risk Factor for Prostate Cancer

Family associated PCa was first reported by Cannon et al. in the Utah Mormon population almost forty years ago in 1982 [19]. Familial aggregation has since been well established and it is now widely accepted that men with a family history of PCa in first-degree relatives are at increased risk of the disease [20,21]. In 2015, Liss et al. reported men with a family history of PCa have a higher incidence of PCa (16.9% vs. 10.8%, *p* < 0.01) and higher PCa specific mortality (0.56% vs. 0.37%, *p* < 0.01) than men without a family history of PCa [22]. It is worth acknowledging that hereditary PCa and familial PCa are distinct. Hereditary PCa can be due to identifiable genetic mutations, such as *BRCA1* or *BRCA2* among others, while familial PCa, a broader term encompassing 15% to 20% of patients, can include patients with a strong family history of PCa but no detectable genetic mutations [13]. More recently, genome wide association studies have identified at least 100 susceptibility loci associated variants with PCa [23]. Individually the contribution of such variants is incremental in adding to PCa risk, however collectively they account for approximately 30% of familial risk [24]. Interestingly, PCa has been identified as having the highest heritability of all cancer subtypes with one large Scandinavian twin study estimating that up to 57% of the individual variation in risk is due to genetic factors [25]. A second study found similarly high estimates of heritability of 42% for PCa [23]. Despite the recognition of this significant heritable component, it has been challenging to identify the genes associated with PCa predisposition [26] and mutations in high-risk genes explain only a small proportion of hereditary PCa families [27]. The genetic heterogeneity of PCa and high population prevalence has meant that despite considerable research, the identification of rare, highly penetrant PCa genes has been extremely challenging [26].

## 4. *BRCA* Mutations in Prostate Cancer

In analyzing PCa genetics, it is critical to distinguish between localized versus high risk and metastatic disease. Firstly, owing to widespread adoption of PSA, the majority of new diagnosis of PCa are localized low-grade disease with excellent prognosis. These are clinically distinct from comparatively fewer diagnoses of advanced metastatic PCa [28], who are recognized to have the potential for poor disease outcomes. Several studies have shown a different genomic/genetic landscape in metastatic Castrate Resistant PCa (mCRPC) compared to localized PCa [29,30]. It is problematic to derive meaningful clinical predictions when examining PCa as a whole given the broad clinicopathological heterogeneity of the disease. This can be illustrated by germline mutations in *BRCA2* having been previously underappreciated as a driver of hereditary PCa. Genomic profiling of PCa was first extrapolated from material acquired at unselected prostatectomies and therefore genetic defects were considered to be rare [31]. Therefore, ascertainment bias has thwarted the reporting of the true prevalence of pathogenic genetic mutations in advanced metastatic PCa.

### 4.1. Prevalence of BRCA Mutations in Prostate Cancer

Advanced disease is associated with higher proportion of germline mutations in DNA-repair genes in patients with PCa, with 4.6% prevalence identified with germline mutations in localized cancer and 11.8% to 16.2% witnessed in the metastatic setting [11]. Pritchard et al. reported observing incidence rates of germline *BRCA2* (5.35%) and *BRCA1* (0.87%) in the advanced PCa setting [11]. Robinson et al. performed genomic profiling on mCRPC biopsy samples from 150 affected individuals with metastatic disease with complete integrative clinical sequencing results (whole exome, matched germline, and transcriptome data). They identified *BRCA2* of both the somatic and pathogenic germline alterations in 19/150 (12.7%) cases. Eight affected individuals (5.3%) harbored pathogenic germline *BRCA2* mutations with a subsequent somatic event that resulted in biallelic loss [29]. These results report are similar incidence of germline *BRCA2* to the Prichard et al. study. Another study observed germline and somatic DNA mutations in 944 men with both early and advanced PCa [32]. The most frequently mutated gene was *BRCA2*, pathogenic variants of which were identified in 11.4% of distinct samples with pathogenic *BRCA1* variants identified at a frequency of 1%. Unfortunately, a limitation of the study was their inability to distinguish between germline and somatic mutations and therefore these figured lack granularity on frequency of germline and somatic mutations. This compares with 1–3% of unselected early PCa’s which harbor *BRCA2* germline mutations [33].

### 4.2. BRCA Mutations in Prostate Cancer by Mutation Type

Dall et al., referenced above, provided further granularity on the nature of these mutations, particularly in *BRCA2* mutation carriers [32]. They examined both germline and somatic mutations from 944 unique men, sequencing samples using Foundation medicine. From the 105 samples identified as having alterations in *BRCA2*, the majority were truncating mutations (70%). Twenty-eight samples (26%) had complete deletion and 3 samples had point mutations. Of the 5 prostate samples with biallelic alterations in *BRCA2*, each allele demonstrated different frequencies. The majority of the 10 samples with identified *BRCA1* alterations (80%) were truncating and all were monoallelic with allele frequencies of 0.4 to 0.8, suggesting a germline mutation. Of note, 60% had coincident TP53 alterations, an occurrence also described in *BRCA1*-associated breast cancers [32]. The varying genetic profile between localized PCa and metastatic lesions may be the result of tumor development beneath the discriminating pressure of prior treatments, such as Androgen Deprivation Therapy (ADT), Androgen Receptor Signaling Inhibitor (ARSi) or chemotherapy. However, it is of course possible that certain mutations might be already present in primary tumors and might expand during the development of metastatic disease and thus become the driver the of disease [6]. The Catalog Of Somatic Mutations in Cancer (COSMIC) reports 312 somatic BRCA2 mutations from 4093 samples tested across all subtypes of PCa, a percentage prevalence of 7% [34]. The breakdown mutation type and observed substitution mutations is demonstrated in Figure 1.

## 5. Testing Strategies for *BRCA* in Prostate Cancer

Since the Human Genome project in 2000, genetic sequencing has become quicker, cheaper and with improved detection of low frequency somatic variants. Yet somatic testing introduces certain challenges. For instance, malignant cells can acquire genetic mutations over time, as well as the divergence of clonal populations of cells during tumor evolution and treatment pressure [35]. While a mutation arising early (truncal), will be present in all the derived cell clones, which arise from the progenitor, mutations that arise later, will only be present in a sub populations of cells. This gives rise to genomic heterogeneity that is absent from germline genomics. Furthermore, cell populations from solid tumors, are frequently mixed with nonmalignant type cells, thus diluting the relative abundance of tumor associated alleles. For these reasons, accurate analysis of cancer genomes typically require two or three times higher sequencing coverage than analysis of germline genomes [36]. Patients should be informed that somatic tumor sequencing has the potential to uncover germline findings with the caveat that virtually none of the NGS tests is designed or validated for germline assessment [28]. If a germline mutation is suspected either because a somatic mutation was discovered on tumor testing that may be of germline origin or there was a clinically suspicious of a germline mutation, the patient should be recommended for genetic counseling and follow-up dedicated germline testing. Conversely, somatic testing also misses a proportion of germline mutations. NCCN has guidelines on when to screen for germline mutations (see below in Section 7) and have recently added some additional guidelines on somatic/tumor testing [28]. While NCCN states a preference for a metastatic biopsy for histologic and molecular evaluation, when not possible plasma ctDNA assay is an option. This is not an uncommon problem in PCa, where accessibility of bony metastatic deposits may compromise yield for sequencing. However, challenges of accurate cfDNA testing are being described [37]. A report which examined cfDNA samples from 40 patients and sent them to 2 separate CLIA-certified laboratories, reported only 9 of 40 (23%) demonstrated congruence (complete or partial) of positive findings [38] highlighting inaccuracies of commercial laboratory testing, therefore vigilance needs to be taken when adopting this strategy.

## 6. Clinical Impact of *BRCA* Mutations in Prostate Cancer

### 6.1. Risk of Prostate Cancer in Those with BRCA Mutations

The probability of developing PCa increases from 0.005% in men younger than 39 years to 2.2% in men between 40 and 59 years and 13.7% in men between 60 and 79 years in the general population [39]. Understanding the risk of PCa in those harboring a germline *BRCA* mutation was initially established from epidemiological studies of breast cancer families as mentioned above. Specifically, retrospective analyses from the Breast Cancer Linkage Consortium showed a relative risk (RR) of PCa in male *BRCA2* and *BRCA1* mutation carriers of 4.65 (95% confidence interval (CI): 3.48–6.22) and 1.07 (95% CI 0.75–1.54) respectively, the latter being similar to the general population. Notably, this study reported that risk increased in younger men, with male *BRCA2* and *BRCA1* carriers under 65 years of age reported as having a the RR of PCa of 7.33 (95% CI 4.66–11.52) and 1.82 (95% CI 1.01–3.29, *p* = 0.05), respectively [40,41,42]. A more recent study screened 1864 men with PCa for *BRCA2* mutations and reported a prevalence of *BRCA2* mutation of 1.20% for cases ≤65 years. They derived an 8.6-fold increased risk of PCa for *BRCA2* mutation carriers by age 65, corresponding to an absolute risk of approximately 15% by age 65 [10]. This is similar to a previous study, which reported the 7.3 fold RR for patients with *BRCA2* mutations to be diagnosed with PCa before age 65 years [41]. Nyberg et al. reported in a prospective analysis that carrying a *BRCA1* mutation was associated with a PCa Standardized Incidence Ratio (SIR) of 2.35 (95% CI 1.43–3.88) relative to the population incidence, whereas the SIR for *BRCA2* carriers was 4.45 (95% CI 2.99–6.61) [43]. They reported increased incidence in PCa in *BRCA1* in younger populations similar to previous studies, which reported an SIR of 3.57 (95% CI 1.68–7.58) for ages <65 year. BRCA2 carriers had an SIR of 3.99 (95% CI 1.88–8.49) for <65 year. The study further reported the estimated absolute risk of PCa was 21% (95% CI 13–34%) by age 75 year and 29% (95% CI 17–45%) by age 85 year for BRCA1 carriers. The corresponding PCa risks for BRCA2 carriers were 27% (95% CI 17–41%) and 60% (95% CI 43–78%) respectively [43] (see Table 2).

A recent meta-analysis performed by Oh et al., has confirmed that being a BRCA mutation carrier (*BRCA1* and/or *BRCA2*) was associated with a significant increase in PCa risk (Odds Ratios (OR) = 1.90, 95% CI = 1.58–2.29), with *BRCA2* mutation being associated with a greater risk of PCa (OR = 2.64, 95% CI = 2.03–3.47) than *BRCA1* (OR = 1.35, 95% CI = 1.03–1.76) [44]. Based on the data from the Kote-Jarai et al. study, the strongest predictors for the presence of a germline *BRCA2* mutation are a young age of onset of PCa and a family history of breast and/or ovarian cancer. Furthermore, no mutations were found in cases diagnosed >65 in their series, suggesting that germline *BRCA2* mutation is far more closely linked to a younger age of PCa onset than to a family history of PCa [10]. Leongamornlert et al. estimated that germline mutations in the *BRCA1* gene confer a RR of PCa of approximately 3.7 fold, which translates to an 8.6% cumulative risk by the age of 65 years [45].

### 6.2. Risk of Prostate Cancer Agressiveness in Those with BRCA Mutations

Not only are men who harbor *BRCA* mutations more likely to get PCa and more likely to get it at a younger age, men with PCa who are carriers of *BRCA2* have more aggressive disease. In fact, a de novo high risk PCa diagnosis prompts germline testing for patients as per NCCN recommendations as expanded on below [46]. Initial studies from Iceland reported that PCa occurring in *BRCA2* mutation carriers were more aggressive than those in non-carriers, and had poorer survival [16] however it was unclear if this may have been due to a specific mutation type (999del5). A further study by Edwards et al. showed that in the UK population, men with a variety of other mutations in the BRCA2 gene, are also likely to have a similarly poor survival [40]. This was mirrored in the Kote-Jarai et al. study where the proportion of high-grade PCa (Gleason score ≥ 8) was 63%, significantly higher than in non-carriers, 15% (*p* < 0.0001) [10]. A Gallagher et al. study reported that *BRCA2* mutation carriers had an increased risk of PCa and a higher histologic grade, and *BRCA1* or *BRCA2* mutations were associated with a more aggressive clinical course [31], findings confirmed by studies by Castro et al. in a large retrospective cohort [47]. Most recently, Pritchard reported *BRCA2* to have the highest relative risk of metastases when compared with men who do not have PCa [11].

In addition to the above studies in Iceland and the UK [16,40], various other international studies have reported the poorer survival associated *BRCA2* mutation carriers [21,40,47]. The median OS of all *BRCA2* mutation carriers was reported by Edwards et al. as significantly shorter at 4.8 years compared with that of non-carriers at 8.5 years (*p* = 0.003) and that poorer survival was associated with the presence of a germline *BRCA2* mutation, which was independent from stage or histology at time of diagnosis [40]. However, Abida et al. conducted a longitudinal analysis of patients with mCRPC harboring genetic alterations, found no association between grouped *BRCA2, BRCA1* and *ATM* altered genes and time on treatment or OS [48]. It is possible that by grouping genes together, the effect of *BRCA2* gene, which is known to have poorer survival compared with BRCA1 is diluted down. One study reporting the median survival time was 8.0 years for BRCA1 carriers and 4.0 years for BRCA2 carriers [49]. This represents a significance difference, with the *BRCA2* gene being associated with a more aggressive phenotype [50]. Conflicting data exists for the prognostic role for *BRCA2, BRCA1,* and *ATM* alterations. One study showed a better prognosis for tumors with alterations in these genes [51] and another study showed no impact for the presence of germline DNA repair gene alterations on outcomes in mCRPC [52]. A recent meta-analysis cited above by Oh et al., reported that OS (HR = 2.21, 95% CI = 1.64–2.30) and cancer-specific survival (CSS) (HR = 2.63, 95% CI = 2.00–3.45) were significantly worse among BRCA2 carriers compared to noncarriers, whereas OS (HR = 0.47, 95% CI = 0.11–1.99) and CSS (HR = 1.07, 95% CI = 0.3–2.96) were not significant when comparing BRCA1 carriers and noncarriers [44].

## 7. Screening/Testing for *BRCA* Mutations in Prostate Cancer

Screening in PCa has been controversial since 1986 when it was widely adopted after its approval by the US Food and Drug Administration (FDA) [53]. There have been diverging views on the utility of the ubiquitously adopted PSA, with major concerns regarding the risk of over-diagnosis and over-treatment [54]. Selecting patients who may benefit from more rigors screening programs such as patients with higher risk disease or a pathogenic driver mutation may alleviate this issue. The results of the IMPACT study (NCT00261456), an international targeted prostate screening study of men at increased PCa risk due to the presence of known pathogenic mutations in *BRCA1* and *BRCA2* genes, are pending and will be of great interest to the topic [55].

In response to the newfound appreciation of the frequency of germline mutations and their implications in PCa, the NCCN has compiled guidelines for clinicians [28]. The panel recommends germline genetic testing for patients with PCa and any of the following; (i) a positive family history of high risk germline mutation; (ii) high-risk, very-high-risk, regional, or metastatic PCa regardless of family history as defined by the guidelines; (iii) Ashkenazi Jewish ancestry (see Section 9.2) (iv) a strong family history of PCa (outlined below) and/or (v) Intraductal (IDC)/Cribriform (CRIB) histology. This association between IDC and CRIB histology is independently associated with bi-allelic BRCA2 alterations with odds ratio(OR) of 4.3 (95%CI 1.1–16.2) and OR of 5.6 (95%CI 1.7–19.3) respectively [56].

A strong family history of PCa consists of brother or father or multiple family members who were diagnosed with PCa (but not clinically localized Grade Group 1) at <60 years of age or who died from PCa, or more than 3 cancers on same side of family, especially diagnoses ≤50 years of age including bile duct, breast, colorectal, endometrial, gastric, kidney, melanoma, ovarian, pancreatic, prostate, small bowel, or urothelial cancer. ESMO guidelines are similar, recommending offering germline screening regardless of tumour features in the advanced setting. Men with localized prostate cancer should also be considered for germline testing if at least two close blood relatives on the same side of the family have been diagnosed with tumors linked to hereditary cancer predisposition syndromes [57].

As per the NCCN germline testing when performed, should include the homologous recombination genes *BRCA2, BRCA1, ATM, PALB2* and *CHEK2* and *MLH1, MSH2, MSH6,* and *PMS2* (for Lynch syndrome). Further, the NCCN recommend that men with a pathogenic or likely pathogenic *BRCA2* or *BRCA1* mutation should start PCa screening at age 45 years. Testing should also be strongly considered by men with a broader family history, including a family history of hereditary breast and ovarian cancer, hereditary PCa or Lynch syndrome [46]. The NCCN reminds clinicians that these guidelines are primarily based on expert opinion and there remains a paucity of strong evidence defining a clinically relevant disease risk which warrants genetic testing.

The Philadelphia Prostate Cancer Consensus Conference guidelines 2019 state that men with metastatic PCa should have germline testing in BRC2/BRCA1, DNA MMR genes (recommended) while *ATM* gene testing should be considered. While Somatic next-generation sequencing for all men with metastatic PCa is recommend. The guidelines specify that additional genes may be considered for testing on the basis of personal or family history [7].

Of note, while multiple guidelines now include germline genetic testing for men with PCa, incorporating these recommendations into clinical workflows remains a challenge [58]. Unfortunately access to genetic providers is often limited. The increased number of men with PCa to be tested and the scarcity of genetic counsellors suggest a critical unmet need for expanded genetic services through novel approaches outside of historic delivery models [59].

## 8. Current Treatment Options for *BRCA* Mutated Prostate Cancer

For the majority men diagnosed with PCa, their outcomes are encouraging; with 98% of men in the USA and 83% in Europe alive 5 years after diagnosis [3]. Early stage PCa can be definitively treated with radiation treatment or surgery, and given the indolent nature of some PCa’s there can be a place for surveillance [60]. PCa will recur in approximately 20% to 30% of unselected men (i.e., unknown mutation status) treated for localized PCa. Since Huggins and Hodges won a Nobel Prize in 1966 for their work describing the relationship between testosterone and PCa, Androgen-Deprivation Therapy (ADT) has continued to be an important component in the treatment of advanced PCa [61]. Despite the majority of men with metastatic PCa at first responding to ADT, the average duration of response in metastatic castration-sensitive PCa (mCSPC) spans from 24 to 36 months. Inevitably their PCa stops responding to this therapy, a condition known as castration-resistant PCa (CRPC) [60]. The average survival for patients with mCRPC is less than 2 years [62] despite new therapeutic strategies for CRPC being offered to patients, such as new combinations and sequences of second-generation antiandrogen therapy (enzalutamide, abiraterone, apalutamide) or second line chemotherapy (cabazitaxel), which have shown notable benefit for patient survival [60].

There are three large phase clinical trials showing efficacy of Poly (ADP-ribose) polymerase (PARP) inhibitors (PARPi) in the advanced setting detailed below, however there is otherwise a lack of high level evidence surrounding the distinct treatment of *BRCA* mutation carriers within the current treatment framework. There is particularly a lack of evidence guiding radical treatment of localized PCa for *BRCA* mutation carriers [63,64]. Building on the progress achieved in the advanced PCa setting, combined with the knowledge that *BRCA* mutant PCa is more aggressive, it is iterative to explore the role of more rigors treatment regimes in the localized high risk setting. This mirrors similar studies which have been performed in the breast cancer space. Recent phase 3 double blinded clinical trial by Tutt et al. demonstrated that among patients with high-risk, early breast cancer with an identified germline *BRCA1* or *BRCA2* mutations, treatment with a PARPi (Olaparib) after completion of local treatment and chemotherapy was associated with significantly longer survival free of invasive or distant disease versus placebo [65]. Clinical trials are underway examining if there is a role for neo-adjuvant or adjuvant treatments with PARPi for patients with a *BRCA* mutation [66,67].

### 8.1. PARPi for BRCA Mutated Prostate Cancer

*In vitro* studies have demonstrated that PCa cells harboring *BRCA1* and *BRCA2* mutants are up to 1000 times more sensitive to PARPi [68,69]. At present there are two PARPi, Olaparib and Rucaparib which have demonstrated efficacy in patients with genetic mutations in PCa, namely *BRCA2*, *BRCA1*, or *ATM* mutations [12,70]. Olaparib has known survival benefit for patients diagnosed with breast cancer or High Grade Serous Ovarian Cancer (HGSOCa) with homologous recombination deficiency [65,71,72,73]. PROfound was a phase III randomized controlled trial comparing a PARPi (Olaparib) with a Androgen Receptor Signaling Inhibitors (ARSi) in men with mCRPC [70]. Men who were eligible were men with mCRPC who’s diseased had not responded or stopped responding to treatment with an ARSi, namely Abiraterone or Enzalutamide. Men with three identified gene alterations (*ATM, BRCA1* and *BRCA2*) were treated with the PARPi or an ARSI (either Abiraterone or Enzalutamide). Another subgroup was formed of men with twelve additional predetermined gene defects, who received a PARPi or an ARSi as above. When the overall population was assessed, the patients treated with PARPi had improved outcomes compared to the ASRi group (PFS 5.8 months vs. 3.5 m; HR, 0.49; 95% CI, 0.38 to 0.63; *p* < 0.001). The advantage was most marked in the first group (PFS 7.4m vs. 3.6m; HR for progression or death, 0.34; 95% (CI), 0.25 to 0.47; *p* < 0.001). Similarly, TRITON2 a phase II open label study, investigated the benefit of Rucaparib in PCa for men with mCRPC with deleterious *BRCA* mutation (germline and/or somatic) previously treated with ARSi and a taxane based chemotherapy. The populations included 115 patients with a *BRCA1* alteration (n = 13) and *BRCA2* alteration (n = 102). Confirmed ORRs were 43.5% (95% CI, 31.0% to 56.7%; 27 of 62 patients) and 50.8% (95% CI, 38.1% to 63.4%; 33 of 65 patients), respectively. The confirmed PSA response rate was 54.8% (95% CI, 45.2% to 64.1%; 63 of 115 patients). ORRs were similar for patients with a *BRCA1* or *BRCA2* alteration, while a higher PSA response rate was observed in patients with a *BRCA2* alteration. The outcomes of this study resulted in its approval by the FDA [12]. There are currently over twenty clinical trials involving PARPi in combination with other treatment modalities listed on clinicaltrials.gov (accessed on 13 November 2021) in the advanced PCa space in an attempt to establish optimum treatment sequence. This has been summarized and tableted by Sigorski et al. [68]. Of note, eligibility for inclusion in clinical trials for Olaparib and Rucaparib required the identification of a pathogenic mutation by either germline and/or somatic testing [12,74].

### 8.2. Chemotherapy for BRCA Mutated Prostate Cancer

BRCA2 mutations have been associated with a higher likelihood of response to carboplatin-based chemotherapy than non-*BRCA2*-associated PCa in CRPC [75], which is somewhat foreseeable given Carboplatin is a standard treatment for *BRCA1* and *BRCA2* patients in ovarian and breast cancer settings. Platinum-based chemotherapy, alkylating DNA, induces genomic strand breaks that may be translated in a synthetic lethality in tumor cells with DNA Damage Repair (DDR) mutations.

At present, only two taxane-based chemotherapies, namely Docetaxel and Cabazitaxel, have shown efficacy in the advanced PCa setting [60]. Gallagher et al. reported half of *BRCA* carriers had a PSA response to taxane-based chemotherapy, suggesting that it is an active therapy in these individuals, with 71% (54/76) of patients responding to treatment, with no significant difference between carriers (57%) and non-carriers (72%) (absolute difference 15%; 95% CI, 23% to 53%; *p* = 0.4) [76]. Trials assessing the efficacy of platinum-based chemotherapies in mCRPC patients with *BRCA* mutations are ongoing [77,78]. Platinum chemotherapy generates interstrand cross-links that can only be adequately repaired by HR based DNA repair, and consequently BRCA1 and *BRCA2* mutated cells are highly sensitive to platinum chemotherapy both in vitro and in vivo [79]. It is also a matter of considerable interest whether men with identified *BRCA* mutation in PCa would benefit from chemotherapy early in their disease course.

### 8.3. Androgen Receptor Signalling Inhibitors for BRCA Mutated Prostate Cancer

There are a number of second-generation antiandrogen therapies (enzalutamide, abiraterone, apalutamide and darlutamide) with proven efficacy in PCa [80,81,82,83]. Antonarakis et al. evaluated the clinical significance of DDR mutations in 172 mCRPC patients receiving first-line ARSi and found that *ATM-BRCA1/2* carriers had a trend towards longer progression-free survival (PFS) than noncarriers (15 vs. 10.8 months, *p* = 0.090) [84]. However Annala et al. reported that defects in *BRCA2* and *ATM* were strongly associated with poor clinical outcomes independently of clinical prognostic factors and circulating tumor DNA abundance when treated with abiraterone or enzalutamide [85]. *BRCA2* and *ATM* carriers had a significant shorter PFS than noncarriers (3.3 vs. 6.2 months, *p* = 0.01) when treated with first-line ARSi. Prospective validation in larger patient cohorts will be required given the conflicting results in the literature.

## 9. *BRCA* as a Biomarker for Prostate Cancer Risk in Special Populations

BRCA variation across ethnicities has been well established across multiple studies in Ashkenazi Jewish, Polish, Icelandic populations and Asian population [86,87,88,89]. The issue of ethnic-specific BRCA variation has important clinical implications [90] in guiding prevention and treatment of BRCA-related cancer. Selected below are ethnic groups where specific founder mutations in BRCA have been established as a significant biomarker for risk for the population group effected.

### 9.1. Icelandic Population

As noted above, Icelandic studies in the 1990s reported that male relatives in breast cancer families were noted to have a 2–3 fold risk of PCa [16]. Initial studies from Iceland reported that PCa occurring in BRCA2 mutation carriers were more aggressive than those in non-carriers, and had poorer survival [16]. It was discovered that a ‘999del5′ mutation in the *BRCA2* gene explains a substantial proportion of familial risk of breast cancer in Iceland. This single *BRCA2* mutation accounts for 7–8% of female breast cancers and 40% of male breast cancers there [91]. The BRCA2 999del5-associated cancer risk is most probably due to haploinsufficiency [92]. Clinical practice guidelines for *BRCA1* and *BRCA2* direct that genetic testing needs to be considered more carefully for women from populations with a small spectrum of founder mutation, with a founder effect, such as in Iceland [93]. Similar founder mutations have been identified in the polish population [94] (see Table 3).

### 9.2. Ashkenazi Jewish Populations

In Ashkenazi Jews, *BRCA* mutations are found in up to 5.2% of unselected patients with PCa [95]. Another study demonstrated that more than 2% of Ashkenazi Jews carry these germline mutations, specifically the mutations in *BRCA1* involving the deletion of an adenine and guanine (185delAG) and the insertion of a cytosine (5382insC), and a mutation in *BRCA2* involving the deletion of a thymine (6174delT). These carriers have a 16% chance (95% CI, 4–30%) of developing PCa by the age of 70 and 39% by the age of 80 [96]. NCCN guidelines reflect this risk by recommending germline testing for any persons of Ashkenazi Jewish descent with a diagnosis of PCa, regardless of local or metastatic status [28]. The Philadelphia Prostate Cancer Consensus Conference guidelines 2019 however made separate recommendations for Ashkenazi Jews with metastatic PCA (castration resistant or castration sensitive disease (recommend genetic testing) versus men with nonmetastatic PCA (consider testing) [7].

### 9.3. Black African Men

An analysis of population-based cancer registries found that the incidence of PCa was higher in black men than in white men [97]. It has also been reported that African-American men have a higher lifetime risk of developing (18.2% vs. 13.3%) and dying from (4.4% vs. 2.4%) PCa compared to Caucasian-American men [98]. While there is a scarcity of evidence, a small study showed that African American men are less likely to habor BRCA mutations when compared to the Pitchard et al. data [11,99]. Further analysis are underway to explain these clinical disparities. Hypothesis include difference in tumor location within the prostate that may reflect different PCa subtypes, differences in gene expression and/or treatment disparities and access to health care [100,101,102].

**Table 3 cancers-13-05697-t003:** BRCA1/2 founder mutations.

BRCA2/BRCA1	Mutation	Population	References
*BRCA1*	185delAG (c.68_69delAG)	Ashkenazi Jewish	Tenner et al. [103]
*BRCA1*	5382insC (c.5266dupC)	Ashkenazi Jewish	
*BRCA2*	6174delT (c.5946delT)	Ashkenazi Jewish	
*BRCA2*	999del5 (999del5-2T)	Icelandic Population	Tulinius et al. [91]
*BRCA1*	5382inC (c.5266dupC)	Polish Population	Kowalik et al. [94]
*BRCA1*	300T > G (c.181T > G)	Polish Population	
*BRCA1*	185delAG (c.68_69delAG)	Polish Population	
*BRCA1*	4153delA (c.4035delA	Polish Population	

## 10. Conclusions

PCa is a heterogeneous and complex disease, yet historically treatment strategies have not embraced personalized medicine. Mutations in *BRCA* genes has prognostic, predictive, and treatment implications for PCa patients and this paper has explored *BRCA* mutations specifically as a biomarker for risk in PCa. Further, a germline mutation can have familial consequences. *BRCA2* mutations in the germline have been shown to be associated with worse survival outcomes. Clinical trials are needed to guide treatment options, timing and sequence, for this subgroup of men with a *BRCA* mutation who develop PCa. It is reasonable to deduce that these patients would require more vigorous treatment combinations in addition to targeted treatment options, however, high level evidence is needed to confirm this. While heterogeneity in the genomic landscape of PCa has become apparent through several comprehensive profiling efforts similar to the above, more research is required to assess the impact of these genomic nuances on clinical outcome.

## Figures and Tables

**Figure 1 cancers-13-05697-f001:**
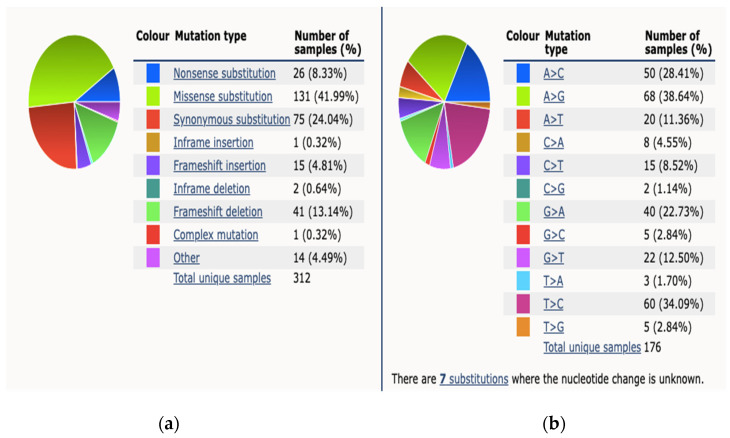
This figure demonstrates mutation distribution for BRCA2 gene as extracted from COSMIC database 2021. (**a**) An overview of the types of mutations observed (**b**) a breakdown of the observed substitution mutations.

**Table 1 cancers-13-05697-t001:** List of identified germline genes that have been implications in advanced prostate cancer from seminal work from Pritchard et al. [11].

Gene	% Prevalence Pathological Gene Mutation in Metastatic PCa
*BRCA2*	5.53%
*CHEK2*	1.87%
*ATM*	1.59%
*BRCA1*	0.87%
*GEN1*	0.46%
*PALB2*	0.43%
*RAD51D*	0.43%
*ATR*	0.29%
*PMS2*	0.29%
*BRIP1*	0.18%
*FAM175A*	0.18%
*MSH2*	0.14%
*MSH6*	0.14%
*RAD51C*	0.14%
*MRE11A*	0.14%

**Table 2 cancers-13-05697-t002:** Tabulated outcomes from prospective analysis by Nyberg et al.

Standardized Incidence Ratios (SIRs)	*BRCA1*	*BRCA2*
SIR relative to population incidence	2.35 (95% CI 1.43–3.88)	4.45 (95% CI 2.99–6.61)
SIR relative to population incidence	3.57 (95% CI 1.68–7.58)	3.99 (95% CI 1.88–8.49)
Absolute Risk of PCa by 75 years	21% (95% CI 13–34%)	27% (95% CI 17–41%)
Absolute Risk of PCa by 85 years	29% (95% CI 17–45%)	60% (95% CI 43–78%)

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
