# Peer review of "Pathogenic *BRCA* Variants as Biomarkers for Risk in Prostate Cancer"

_cancers, 2021, doi:10.3390/cancers13225697_

Round 1
Reviewer 1 Report
This is a very well written and informative review. The topic is of great interest as the field is progressing at a rapid pace. Overall, the broad perspective into the origins of this research avenue, and importance of leaning from the experience in breast and ovarian cancer is insightful and relevant. The major studies in the field are cited and clinical development is also current. A strength of the review is the detailed reference to the current guidelines from NCCN and ESMO, which provides standard practice information. My suggestions for consideration are as follows:
- Consider making a clearer statement to distinguish between germline and somatic mutations early in the manuscript. While the beginning of the article refers to germline mutations, the inclusion of somatic mutations in section 4.1 is not clearly articulated and can be confusing.
- Line 124: Is this 12.7% somatic mutations?
- Line 127: germline and somatic mutations are included, are the stated results combined or for each type?
- The text at the end of the introduction (lines 47-54) is repetitive with section 6.1 (lines 179-218) and the information provided there is included in Table 2 in the same section. Consider combining.
- The point about having limited information about treatment for men with localised prostate cancer with known BRCA2 mutations is well made. Consider making this point more prominent, and include ref PMID: 30808988.
- The clinical context of prostate cancer is presented in section 4 (lines 102-115), which is helpful. I would recommend being more specific about the definition of ‘advanced’. My understanding is that you are referring to ‘high-risk’ prostate cancer with this term (i.e. high-grade localised disease), but it can be confused with metastasis. Please clarify.
- The inclusion of the Dall study is slightly repetitive in section 4.1 and then 4.2 where the same 944 patient cohort is described. Consider revising the way the studies are described (Line 127-135).
- I would recommend including more detail on the knowledge on pathology in BRCA2 prostate cancer. IDCP has been reported as a prevalent pathology in these patients, and this is only briefly mentioned in the NCCN guidelines, but not mentioned elsewhere (line 273). This is important co-risk factor that has clinical relevance. A recent study by Lozano addresses this concept (PMID: 33626496).
- Consider revising the order of the information given around PARPi’s in prostate cancer. The main point is that these therapies were FDA approved for prostate cancer treatment in 2019/2020, yet this is not stated until line 356. This is important to have upfront in section section 8; as it reads, it is approved in breast cancer but not yet in prostate cancer until later in the paragraph.
Minor points:
- Line 55: Table 1. Include reference to this being ‘germline’ or ‘inherited’ genes in the title.
- Line 53: Consider removing reference to ‘scientific’ in the introduction (line 53) as there is very little biology included.
- Line 77: Consider including PMID: 21733824 with refs 19,20, when discussing family history from breast to prostate cancer.
- Line 81: delete repeated reference.
- Line 132: consider revising title. As it reads, it could imply that you will talk about different types of prostate cancer, rather than types of BRCA mutations.
- Line 133: change BRCA2 positive patients to BRCA2 mutation carriers or the like.
Author Response
Please find the author's response attached below.

Reviewer 2 Report
This review is about the prognostic, clinical, and therapeutic role of BRCA1/2 gene in prostate cancer. Overall, this review seems to be well organized by referring to important matters. However, I have several concerns for this review.
- Recently, several reviews related to association between prostate cancer and BRCA1/2 including other genes or current treatment/clinical trial for prostate cancer have been published (PMID: 32790034, 32963528. 30730411, 32516092, 33044685, etc.). Therefore, to ensure the novelty, please include a sentence in the abstract and again in the introduction or conclusion how this is different than what have already been published.
- This review was explained too descriptively throughout the entire text. Therefore, making it somewhat less readable, and it seems that more effective arrangement using tables or figures is needed for readers to understand more easily.
- It would be better to introduce the recommendations by the 2019 Philadelphia Prostate Cancer Consensus (PMID: 32516092) along with germline genetic testing proposed by the NCCN guidelines in the 7. Screening/testing for BRCA mutations in prostate cancer.
- Many studies have been conducted on BRCA1/2 pathogenic variants by ethnicity/country, and there are already studies summarizing them (PMID:29446198, 32963034, etc.). These studies showed that BRCA1/2 foundation pathogenic variants were different by regions/ethnicities. Therefore, it would be better if these contents were added to 9. BRCA as a biomarker for prostate cancer risk in special populations.
Author Response
Please find the author's response in the attached file below.

Reviewer 3 Report
The data used belongs to Pritchard et al prior to 2016, Robinson et al prior to 2015, even to make Table 1. This is old and limited data without completing with other authors. At no time is the COSMIC (Catalog of somatic mutations in cancer) used, where mutations are collected from more than 4500 prostate carcinomas and the last update is from 2021.
Author Response

(The authors gave the same response as above.)

Round 2
Reviewer 1 Report
Thank you to the authors for making the suggested amendments.
The inclusion of additional details on the prevalence of IDCP and BRCA2 mutations is important, and I would recommend including addition studies in germline BRCA2 mutation carriers, such as PMID: 25154392 (lines 282-284).
Otherwise, all comments have been satisfactorily addressed.
Author Response
Please find our responses attached below. Thank you.

Reviewer 2 Report
They have addressed all comments and concerns to my satisfaction. It's a minor mistake, but please write the gene in italics.
Author Response

(The authors gave the same response as above.)

Reviewer 3 Report
Authors have answered perfectly to all the comments and have clarified some sections in the text.
Minor: Figure 1 should be adapted to the manuscript format.
Author Response

(The authors gave the same response as above.)
